# Bistable Bacterial Growth Dynamics in the Presence of Antimicrobial Agents

**DOI:** 10.3390/antibiotics10010087

**Published:** 2021-01-18

**Authors:** Nelly Frenkel, Ron Saar Dover, Eve Titon, Yechiel Shai, Vered Rom-Kedar

**Affiliations:** 1Department of Biomolecular Sciences, 206 Nella and Leon Benoziyo Building for Biological Sciences, The Weizmann Institute of Science, Rehovot 76100, Israel; nellyfre@gmail.com (N.F.); rsdover@stanford.edu (R.S.D.); 2Department of Computer Science and Applied Mathematics, 211 Jacob Ziskind Building, The Weizmann Institute of Science, Rehovot 76100, Israel; eve.titon.research@gmail.com

**Keywords:** microbiology, antibiotics, antimicrobial peptides, inoculum effect, mathematical modeling of infection

## Abstract

The outcome of an antibiotic treatment on the growth capacity of bacteria is largely dependent on the initial population size (Inoculum Effect). We characterized and built a model of this effect in *E. coli* cultures using a large variety of antimicrobials, including conventional antibiotics, and for the first time, cationic antimicrobial peptides (CAMPs). Our results show that all classes of antimicrobial drugs induce an inoculum effect, which, as we explain, implies that the dynamic is bistable: For a range of anti-microbial densities, a very small inoculum decays whereas a larger inoculum grows, and the threshold inoculum depends on the drug concentration. We characterized three distinct classes of drug-induced bistable growth dynamics and demonstrate that in rich medium, CAMPs correspond to the simplest class, bacteriostatic antibiotics to the second class, and all other traditional antibiotics to the third, more complex class. These findings provide a unifying universal framework for describing the dynamics of the inoculum effect induced by antimicrobials with inherently different killing mechanisms.

## 1. Introduction

Antibiotic resistance is rising to dangerously high levels in all parts of the world. Over-prescription and misusage cause natural selection that favors strains with new resistance mechanisms, threatening our ability to treat common infectious diseases effectively. Many factors can affect bacterial susceptibility to antibiotics, including the metabolic state and the presence of persistent cells [1,2,3], the microenvironment conditions that affect the antibiotic potency [4], the physical structure of the population (biofilms) [5], and the population size, or inoculum, at the site of infection. Indeed, a major power of bacteria is within numbers since it has been well established that as a population, bacteria often survive a concentration of an antimicrobial agent that is lethal to individual cells. In a therapeutic context, this means that the fate of an initial infection depends on the initial load of bacteria—while small infections are easily cleared even with no antibiotics, large infections are hazardous, even when antibiotics are administered at high doses. This phenomenon, known as the “inoculum effect” (IE), is well established in-vitro [6,7,8,9,10,11,12], as well as in-vivo in animal models and in human patients [11,13,14,15,16,17,18,19].

The IE is a bistable effect [9,12]—a system is said to be monostable if it always equilibrates to one final state, and bistable when it admits more than one stable state [20]. In a clinical situation, having bistable behavior is unfavorable since the treatment outcome is difficult to predict even if the dynamics (drug–target interaction) is assumed to be deterministic. Treating with insufficient antibiotics doses can lead to ineffective bacterial clearing and patients mortality [21], as well as favors the selection of drug-resistant strains [22,23]. It is therefore a common practice to administer antibiotics in excessive dosages, which may weaken the immune system. An optimal treatment achieves a complete eradication of the infection with minimal antibiotics dosing, yet, the IE complicates the determination of such an optimal strategy [12,23,24,25,26].

There are several known biological mechanisms that were proposed to account for the IE. First, it is known that the *E. coli* spontaneous beneficial mutation rate is 10^−5^ mutations per genome per generation [27]. Thus, bacterial populations equal to or larger than 10^5^ may contain *genetic heterogeneity* to antibiotic resistance [28,29]. *Phenotypic heterogeneity* [1,2,3,7] is also more prominent in these larger initial populations. This heterogeneity can generate bistability: at low numbers we expect that only one, non-resistant population exists, hence treatment leads to extinction, whereas at high initial numbers heterogeneity may allow a resistant strain to grow, leading to bistability.

Second, *density-dependent mechanisms* may also lead to an IE. The bacterial density affects both the cellular state of the cell, and its interactions with the antimicrobial agent. Cellular communication that is sensitive to population density is called quorum sensing. Quorum sensing enables bacteria to synchronize gene expression and alter their properties to become more resistant to different antibiotics. At low cell densities, a large proportion of the signaling factors disperse before they can be used, and so their production provides a small direct or indirect fitness benefit. At high cell densities, a much greater proportion of the signaling factors is available per cell, and consequently bacteria can better cope with the relevant stressor [30]. Another density-dependent mechanism, which does not involve cellular signaling, is reduction in the antibiotic concentration. For instance, *E. coli* secrete the β-lactamase enzyme that cleaves and inactivates β-lactam antibiotics [31]. A large population produces more β-lactamase then a small population and can therefore degrade the antibiotic faster [32,33,34]. If the population survives and grows, more enzyme is produced and cleaves the antibiotic in a higher rate—a positive feedback mechanism. A similar concept applies when an antibiotic agent binds to its target or to non-specific cell components and its effective concentration is therefore reduced. This might not have much effect when the antibiotic concentration is sufficiently large compared to the population size, but it is expected to lead to a bistable situation at a critical ratio. Even if the antibiotic concentration does not change in time, bistable behavior can arise when growth is proportional to the amount of antibiotic molecules available to each bacterium at the time of exposure [11,35]. The above density dependent mechanisms are inherently nonlinear since they do not increase gradually, proportionally to the bacterial load and antimicrobial concentration, but rather have either a threshold or limiting effects.

In the bacteria-antimicrobial context, a monostable system corresponds to the case where any initial number of bacteria (*B*) reaches the same maximal population size if treated with antibiotic concentration (*A*) that does not kill all cells (“sub-lethal treatment”). A lethal treatment leads to eradication of all loads, and therefore, the fate of the system is independent of the initial load (Figure 1A). In a bistable situation, the fate of the system may depend on its initial load. Here, for a range of the antimicrobial agent concentrations, the bacterial population can assume two possible states (the two solid black lines in Figure 1B), depending on whether the initial bacterial concentration was above or under the threshold concentration${\;B}_{c}\left(A\right)$ (dashed black line in Figure 1B). Three main regimes govern this bistable dependence. First, a minimal antibiotic/peptide concentration $A_{c}$ is needed to kill or inhibit a minute number of bacterial cells, so for $A<A_{c}$, even minute bacterial populations grow, similar to untreated cells. Second, an enormous amount of antimicrobial agent inhibits practically any number of cells, so there exists $A_{e}$ such that for $A>A_{e}$ all relevant bacterial densities are growth-inhibited or killed. Third, for $A_{c}<A<A_{e}$, the greater the concentration of the antimicrobial agent, the larger the bacterial concentration it can inhibit and therefore the threshold concentration $B_{c}\left(A\right)$ is monotonically increasing with A in this range. It follows that for any given antimicrobial agent concentration in the range$\;A_{c}<A<A_{e}$, the fate of the bacterial population depends on whether its initial concentration is above or below $B_{c}\left(A\right)$ (dotted bottom line, Figure 1B). We thus have a bistable behaviour for all $A_{c}<A<A_{e}$ (Figure 1B).

The monostable/bistable behaviors appear in nonlinear models depending on a single dynamic variable—the varying bacterial concentration, inhibited by a fixed control parameter—the antimicrobial agent. The latter determines whether a monostable or bistable behavior ensues (see Malka et al. [20,36], where neutrophils play the role of the antimicrobial agent). Notably, other more complex mechanisms may also result in a bistable behavior. By Ockham’s razor principle, since the concentration-dependent bistable mechanism is the simplest adequate model, and such a model involves hardly any assumptions on the specifics of the antimicrobial-bacteria interactions, it provides the main underlying mechanism for in-vitro bacteria-antimicrobial dynamics when the antimicrobial action remains fixed (and possibly for in-vivo dynamics under specific conditions, e.g., neutropenic conditions in the case of neutrophils-bacteria interactions [20,36]). The inoculum effect [11,34,35] suggests that a similar underlying mechanism may apply to in-vitro bacteria-antibiotics interactions.

Here, we show that every tested antimicrobial agent showed an inoculum effect for *E.coli* cells, regardless of the distinct type and mechanism of action of the antimicrobial agent. In particular, we show, for the first time to the best of our knowledge, that classical antimicrobial peptides that do not have specific targets on the bacterial membrane induce an inoculum effect (this was previously observed for some non-classical antimicrobial peptides [37]). Moreover, we show that the killing induced by these peptides may be described by an even simpler model than those considered in [11,20,35,36]; Here, the abrupt kill of the bacteria by the peptides exhibits IE. After this, the surviving bacteria grow with no influence of the peptides.

## 2. Experimental Materials and Methods

### 2.1. Commercial Antibiotics

Ampicillin sodium salt, kanamycin sulphate, chloramphenicol, carbenicillin disodium salt, oxacillin sodium salt, and gentamicin solution were purchased from Sigma-Aldrich(Rehovot, Israel). Polymixin B was purchased from Fluka BioChemika (Seelze, Germany). and tetracycline hydrochloride was purchased from Sigma-Aldrich (Rehovot, Israel).

### 2.2. Peptide Synthesis and Purification

Peptides were synthesized by using a 433A synthesizer (Applied Biosystems Rhenium Modiin, Israel) on rink amide 0.65 mmol/mg MBHA (4-Methylbenzhydrylamine) resin, using Fmoc protected amino acids. The synthesized products were washed thoroughly with DMF (Di-Methyl-Formamide) and DCM (Dichloromethane), then dried and cleaved. Cleavage was performed by addition of 95% trifluoroacetic acid (TFA), 2.5% water, and 2.5% TIS (Titanium(II) Sulphide). The peptides were then purified (>98% homogeneity) by reverse-phase High Performance Liquid Chromatography HPLC (RP-HPLC). Purification was performed using a C18 column and a linear gradient (Melittin—20–80%, K6L9—10–90%, MSI-78 (Pexiganin) 10–90%) of acetonitrile in water (final fluid containing 0.1% TFA (*v/v*)) for 40 min.

### 2.3. Medium

Initially, we tried a BM2 minimal medium [62 mM potassium phosphate buffer, pH 7.0, 7 mM ammonium sulfate, 1 mM magnesium sulfate, 10 μM ferrous sulfate, 0.5% (*w*/*v*) casamino acids]. We prepared all medium components without glucose, then autoclaved it (sterilize by heating) and then added filtered glucose solution to 0.2% final concentration. Realizing that the minimal medium results in complex behaviour of the control (see discussion of Figure 2 in results), we used the Luria Broth (LB) medium in all the other experiments as described next.

### 2.4. Bacterial Strains

*E. coli* MG1655 with a Lux-kanamycin resistance plasmid (described previously in [38]) was used for generation of all bacterial growth curves except for the curves generated with kanamycin where non-resistant *E. coli* MG1655 was used. Bacteria were grown in minimal medium, Luria Broth (LB) medium or LB+kanamycin (30 µg/mL) medium, shaking in a standard aerated 15 mL tube for bacterial growth at a 37 °C, 200–250 rpm shaking hot room for 4 h (OD_600_ = 1–1.5), and then diluted for the experiments.

### 2.5. Generation of Growth Curves from Various Initial Inoculums

A sterile 96-well plate with a flat bottom was prepared with serial dilutions of the necessary antibiotic/peptide in LB medium (total volume in well 100 µL). In short, an initial high concentration stock of the antibiotic/peptide in DDW was used to make an initial stock in LB to fill 200 µL into wells in row A. Serial dilutions were then made by a multipipette into the other wells already containing 100 µL of LB (the last row was left without antibiotics/peptide as a positive control). Bacteria from a shaking culture as described above were then measured for their OD_600_ and normalized in 2 mL LB medium to an OD_600_ of 0.01 (order of 10^6^ colony forming units (CFU)/mL as measured from live counts). Serial dilutions of the bacteria were then made in additional 5 LB tubes (200 µL from previous dilution were transferred into 1800 µL of LB every time after vortexing). One hundred microliters of the highest bacterial dilution were inserted into wells A1-B12, 100 µL of the second highest bacterial dilution into wells C1-D12, etc. The smallest inoculum contains about 5 CFU and, in all experiments’ controls, exhibited growth (was not empty). The prepared plate was placed in an automatic microplate plate-reader for 16 h with medium shaking speed and an OD_600_ measurement every 20 min.

### 2.6. Antibacterial Activity

The base minimal inhibitory concentration (bMIC) was determined for each antibiotic/antimicrobial peptide (AMP) based on the above-described growth curves—bacterial populations with an initial inoculum of 10^6^ CFU/mL (2 × 10^5^ CFU per well) that finished with an OD_600_ lower than a cut-off of 0.35 after a 16-h incubation were considered as extinct or inhibited. The lowest antibiotic/peptide concentration for which both duplicates went extinct is the bMIC. Similarly, density-dependent MICs were generated for lower inoculums using the same cut-off.

### 2.7. Preparation of Spent Media with Polymixin B (PMB) and Ampicillin (Amp)

Appropriate concentrations of PMB and Amp were prepared in LB medium for subsequent dilutions for an MIC assay described above. *E. coli* MG1655 were grown in shaking LB until an OD_600_ of about 0.4–0.6 and diluted into some of the prepared PMB and Amp stocks to an OD_600_ of 0.1. All types of these media and LB alone were then incubated in shaking in 37 °C for 1 h. Incubated media were centrifuged, and the supernatant was used for a standard MIC assay described above.

### 2.8. Determination of Antibiotic Potency over Time

*E. coli* MG1655 were grown in shaking LB overnight (ON) and then centrifuged for 5 min in 3000 rpm and diluted to an OD_600_ of 1. ampicillin, chloramphenicol, carbenicillin, and tetracycline were added to appropriate concentrations for final MIC determination as indicated in the results to tubes containing either LB or LB and bacteria in an OD_600_ of 1. Some LB tubes with bacteria in the same density were also incubated with rest without any antibiotics. All tubes were incubated for 24 h in shaking and 37 °C. In addition, another inoculum of *E. coli* MG1655 was grown in shaking LB ON. After 23 h, the whole process was repeated for the 1 h samples with the new ON culture. When 24 h had passed, all samples were filtered through 0.2 µm syringe filters and appropriate amounts of all corresponding antibiotics were added to the samples that were incubated with bacteria only (without antibiotics). Finally, all samples were diluted by a factor of 2 with fresh LB for the highest antibiotic concentration and by increasing dilutions for a standard MIC assay as described above. In addition, fresh LB with fresh antibiotics was prepared for each antibiotic for a parallel classic MIC assay as a control by the protocol described above. All MIC assays were performed in 96-well plates over 16h of incubation in shaking (250 rpm) and 37 °C. Final results were determined by the sight of turbidity and OD_600_ measurements.

## 3. Mathematical Models

Three types of deterministic mathematical models are proposed, all admitting bistable behavior as described in the introduction section; the new bistable immediate kill and then A-independent dynamics model (simplest, chart flow 1), the classical A-dependent dynamics model (simple, chart flow 2), and a classical multiple time-dependent factors A-dependent dynamics model (the most complex model presented here, chart flows 2 and 3). Notably, more complex models taking into account detailed molecular processes, stochastic effects, and/or several phenotypic populations exhibiting bistable behavior may be introduced. One of the main results here is that the two simpler models are adequate for describing the data for some of the antimicrobial agents.

We denote by B(t) (bacterial cells/mL) the concentration of the bacteria at time t (hours) and by A the initial concentration of the antibiotics/peptide (µM). The growth curves under the influence of the antibiotics/peptides reflect the overall population growth, which is decreased/delayed by either killing of substantial portions of the population (peptides or bactericidal antibiotics) or by growth-inhibition (bacteriostatic antibiotics).

### 3.1. The Bacterial Growth Model

The simplest bacterial limited growth model is of the form:(1)$$\frac{dB}{dt}=F\left(B\right),\;t>0.$$

Such a deterministic one-dimensional mathematical model is expected to be valid for sufficiently large inoculum, for which stochastic fluctuations leading to extinction are non-probable and when the environmental conditions are fixed (e.g., the nutrients supply hardly changes during the experiment). Several analytical forms for $F\left(B\right)$, such as logistic, Gompertz, and others are commonly used [9,39]. Here we propose that the growth function, $F\left(B\right)$, can be experimentally found by plotting the change in the bacterial inoculum per time interval $\frac{dB}{dt}\sim \frac{B\left(t+dt\right)-B\left(t\right)}{dt}$ versus $B\left(t\right)$. If, to a good enough approximation, a unique curve is found when this procedure is repeated for several initial inoculum $B\left(0\right)$, then model (1) is adequate for describing the control bacterial dynamics. Otherwise, a more complex model is needed. A threshold condition on the deviations of the experimental $\frac{dB}{dt}$ curve from its mean can be set to distinguish between these behaviors.



**Flow Chart 1. The bistable kill model (BIK) dynamics.**

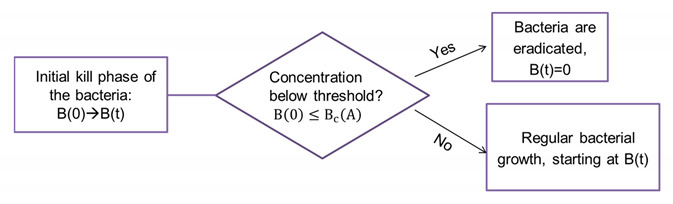




### 3.2. The Bistable Immediate Kill Model (BIK)

The model describes a scenario where the bacterial destruction occurs rapidly at some initial kill phase τ, after which the surviving bacteria, if such exist, grow as if there are no peptides in the medium, namely, as in the control, see flow chart No 1. The mathematical description of this model includes the control bacterial growth function, F(*B*), as above, and a nonlinear kill function,$\;K\left(B\left(0\right),A\right)$, so that $B\left(\tau \right)=K\left(B\left(0\right),A\right)$ is the bacteria concentration after the initial kill phase. Here, the bacterial growth is descried by the equations:(2)$$B\left(\tau \right)=K\left(B\left(0\right),A\right),\;\frac{dB}{dt}=F\left(B\right),\;t>\tau .$$

Such a model *predicts* that if there is bacterial growth, then, if one plots the change in the bacterial inoculum per time interval $\frac{dB}{dt}\sim \frac{B\left(t+dt\right)-B\left(t\right)}{dt}$ versus $B\left(t\right)$ for any initial inoculum and any concentrations of the antimicrobial peptide, a *unique bacterial growth function,* which is identical to the control growth function, F(*B*), will emerge. Here, the bistability is due to the nonlinear threshold form of the kill function.



**Flow Chart 2. The A-dependent dynamics model (BAD)**

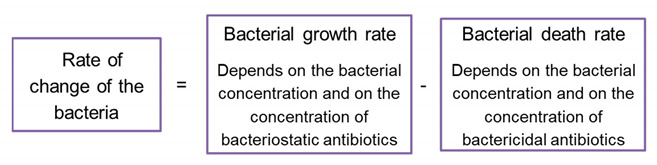




### 3.3. The A-Dependent Dynamics Model (BAD)

The model describes a scenario by which the bacterial destruction or growth inhibition occurs continuously due to the fixed–in-time antibiotics action (See flow chart No 2). The mathematical description of this model is given by a concentration dependent growth function, $F_{BAD}\left(B,A\right)\;$, controlling the bacterial growth under a given antibiotics concentration:(3)$$\frac{dB}{dt}=F_{BAD}\left(B,A\right),\;t>0.$$

Such a model *predicts* that if there is bacterial growth, then, if one plots the bacterial growth function (namely $\frac{dB}{dt}$ versus $B\left(t\right)$) for different initial inoculum and a fixed antibiotics concentration, a unique growth function $F_{BAD}\left(B,A\right),\;$will emerge (yet *different antibiotics concentrations may result in different curves*). Depending on the properties of the function $F_{BAD}\left(B,A\right)$, bistability may emerge (as in Figure 1B, see [20,36]). Then, the MIC increases with the initial inoculum. In [9], the analytic expressions for the density-dependent MIC of six different forms of $F_{BAD}\left(B,A\right)$ are compared to the experimental density dependent MICs. Here we find the experimentally derived $F_{BAD}\left(B,A\right)$ from which the resulting density dependent MIC s derived.



**Flow Chart 3. A bistable multiple factors dynamics model (BMFD) (coupled to flow chart 2).**

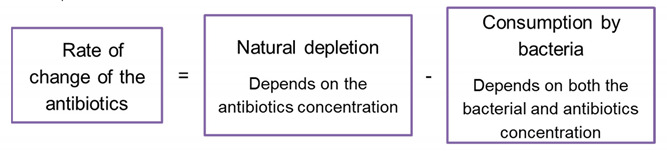




### 3.4. The Bistable Multiple Factors A-Dependent Dynamics Models (BMFD)

These models describe scenarios by which the bacterial destruction or growth inhibition depend on additional factors that vary during the experiment, such as antibiotics and β-lactamase concentrations. Mathematically, such models require *at least* a two-*dimensional* deterministic system, see for example [11,35] (it is also plausible that stochastic terms need to be included, see e.g., [26]). The simplest possible model of this kind is defined by two functions $F_{BMFD}\left(B,A\right),G_{BMFD}\left(B,A\right)\;$that describe the bacterial growth and the depletion of the antibiotics’ concentration by natural decay and by the bacterial dynamics—see flow chart No 3:(4)$$\frac{dB}{dt}=F_{BMFD}\left(B,A\right),\;\frac{dA}{dt}=G_{BMFD}\left(B,A\right),\;t>0$$

Such higher-dimensional models are more complex than the BIK and BAD models. Indeed, these can be considered as two limiting cases of the two-dimensional model (4): Very fast depletion of the antibiotics leads to a BIK model, whereas very slow depletion and consumption of the antibiotics means that the antibiotic concentration remains essentially fixed and the BAD model emerges. *When the experimental growth function for each antibiotic concentration depends on the bacterial initial load, one concludes that the BIK and BAD models are insufficient to describe the bacterial* growth dynamics, so a BMFD model is needed (we do not attempt to fit such models here).

### 3.5. Experimental Growth Functions

To find the bacterial growth functions experimentally, we process their OD_600_ counts as follows. First, to avoid noisy oscillations, we cut the signal at the detection level and after the first maxima and smooth the signal by a standard MatLab algorithm, see supplement for details. Then, we plot $\frac{OD\left(t+dt\right)-OD\left(t\right)}{dt}\sim \frac{dOD}{dt}$ versus *OD*(*t*), thus obtaining the OD growth function. Since the OD is, to a good approximation, linear in the CFU count, for inoculum above the threshold level of 10^6^ CFU/mL, the growth functions, *F*(*B*,*A*), correspond to a linear scaling and shift of the OD growth function graphs.

## 4. Results

### 4.1. A One-Dimensional Mathematical Model Is Adequate to Describe the Bacterial Growth in Rich Medium but not in Minimal Medium

To investigate how the initial bacterial concentration (i.e., initial load) influences the growth outcome at a given antibacterial agent concentration, we first searched for conditions in which the medium content itself is not a limiting factor that can affect the growth dynamics. Six initial bacterial loads with a fold-change between them (roughly 10–10^6^ CFU/mL) were prepared from an exponentially growing culture of *E. coli* and grown in a rich medium (LB) or in a minimal medium (see methods). In the rich medium, all initial loads had the same growth rate during the exponential phase, and reached the same maximal concentration (optical density, OD)—the full capacity of the well (Figure 2A). When grown in a minimal medium, the different initial loads did not necessarily reach the same maximal concentration, nor grew with similar rates during the exponential growth phase. In addition, there was a higher variability between replicated samples (Figure 2D).

Plotting the experimental growth functions, d*OD*/dt versus OD, for the rich and minimal medium experiments, we obtain that for the rich medium all curves converge approximately to a single line (Figure 2B), whereas for the minimal medium they do not (Figure 2D). Namely, in rich medium, the rate of change of the bacterial concentration (d*B*/dt) depends only on its current concentration (*B(t)*) and not on the initial load *B(0)* or the time of growth. Mathematically, such a behavior corresponds to classical deterministic 1D dynamics, so, this verifies that the bacterial dynamics in rich medium is well approximated by the one-dimensional model (1) for at least 16 h.

### 4.2. Bacterial Growth with All Tested Cationic Antimicrobial Peptides Is Bistable and a BIK Mathematical Model Adequately Describes This Growth

Cationic antimicrobial peptides (CAMPs) directly kill bacteria by perforating or completely disintegrating their membrane [40]. We first tested polymixin B, a cyclic peptide derived from the bacterium *Bacillus polymyxa* used in the clinic to fight resistant Gram-negative infections. Figure 3A presents the growth curves for increasing values of peptide concentrations. The panels with *A* = 0.2, 0.3, 0.6 µM of Figure 3A show that while large initial bacterial loads overcame these concentrations of polymixin B, small initial loads could not. This proves that the antimicrobial peptide polymixin B induces a bistable inoculum effect.

Whenever there was growth, all loads at any peptide concentration exhibited the same growth function as the control (see d*OD*/dt curves in Figure 3B and in Appendix A for additional repeats of the experiment). Namely, when there is growth, we again see that the rate of change of the bacterial concentration (dB/dt) depends only on its current concentration (*B(t)*) and not on the initial load *B(0)* or the time of growth. As the CAMP loses much of its potency within an hour (Appendix A), such a behavior is well described by a BIK mathematical model (Equation (2)): The initial load and the peptide concentration determine if growth occurs (via the kill function), and, if growth occurred, it follows the control dynamics.

In Figure 3C, we present the dependence of the normalized maximal capacity of loads that reached their maximum within the experiment for three repeats of the same experiment. This plot corresponds to an experimental bifurcation diagram—the solid lines of the bifurcation diagram of Figure 1 are the asymptotic bacterial loads for a given anti-microbial concentration. While for the BAD model this maximal load decreases with A (see Figure 1), for the BIK model it is A-independent (Figure 3E). The repeats revealed that the bistable range of polymixin B for inoculums of 10–10^6^ CFU/mL is at least 3-fold (0.04≤A≤0.625 µM, see Figure 3C).

Interestingly, the same dynamics were observed for all three other tested antimicrobial peptides—K6L9, MSI, and melittin (Figure 3D and supplementary Appendix A): The bacteria are either completely extinct, or are fully grown by the same growth function as the control, to the maximal capacity of the control well. We thus find that the bistable BIK mathematical model is adequate for describing the growth dynamics in all four tested antimicrobial peptides.

### 4.3. Bacterial Growth with Either Tetracycline or Chloramphenicol Antibiotics Is Bistable, and a BAD Mathematical Model Adequately Describes This Growth

Figure 4A shows the growth curves for different initial loads with increasing concentrations of tetracycline, a bacteriostatic antibiotic that inhibits protein synthesis. Figure 4A demonstrates that bistable behavior appears for 0.8<A<2 µM; at A=1 µM initial loads of 10 CFU/mL do not grow whereas all initial loads greater or equal to 10^2^ CFU/mL grow. At A=1.5 µM initial loads of 10, 10^2^, 10^3^ CFU/mL do not grow and the ones with loads 10^4^, 10^5^, 10^6^ CFU/mL do grow, see the respective panels of Figure 4A. Thus, bistability is verified for this range.

Figure 4B shows that for a given *A,* all loads grow with a similar growth function: Again, the rate of change of the bacterial concentration (*dB/dt*) depends only on *A* and on its current concentration (*B(t)*) and not on the initial load *B(0)* or the time of growth. Mathematically, such a behaviour corresponds to classical deterministic 1D dynamics with a growth function that depends on *A*. Thus, the BAD model (2) is an adequate mathematical model for describing the bacterial dynamics in these experiments. Note that the simpler BIK model cannot describe this behavior. Surprisingly, the experimentally derived function $F_{BAD}\left(B,A\right)$ is not unimodal for A≥0.3 µM, namely there are two loads at which local maximal growth occurs.

Figure 4C shows how the normalized maximal capacity of loads that reached their maximum within the experiment depends on the tetracycline concentration. As predicted by the BAD mathematical model, the maximal capacity depends only on the tetracycline concentration (and not on the initial inoculum), and, as in Figure 1, the maximal capacity decreases with the tetracycline concentration (here, by as much as 20%).

Figure 4D shows that the bistability result and the dependence of the normalized maximal capacity on the tetracycline concentration are robust. Repeating the same experiment several times (6), bistability is observed when the bistable concentration range is sampled (Appendix A show the individual experiments). Moreover, even though the maximal capacity of the control changes considerably between experiments (see Appendix A) the dependence of the normalized maximal capacity on the tetracycline concentration remains essentially unchanged.

The same type of results is obtained with chloramphenicol (the other bacteriostatic agent we tested), with bistability emerging at different concentrations, a bit noisier dependence on the experimental conditions, and with much stronger dependence of the maximal capacity on the chloramphenicol concentration (Appendix A).

### 4.4. Bacterial Growth with Either Gentamycin, Kanamycin, Ampicillin, or Carbenicillin Antibiotics Is Bistable. At the Bistable Range the Bacterial Dynamics Is Complex and Cannot Be Adequately Described by a BIK nor a BAD Mathematical Model

The aminoglycoside bactericidal antibiotics gentamycin and kanamycin bind to the 30S subunit of the bacterial ribosome and inhibit translation, which eventually causes cell death. We find that at a fixed kanamycin concentration in the bistable range 3.75≤A≤15 µM, starting with initial loads of 10–10^6^ CFU/mL, the loads that grow do not grow to the same maximal concentration (Figure 5A), and the specific growth rates of the different initial loads do not overlap (Figure 5B). Thus, no growth functions can adequately describe the dynamics by a BAD or a BIK mathematical model. Thus, at the bistable range, the normalized maximal capacities are different for different loads (Figure 5C). Describing the dynamics requires a more complex model (a BFMD). Figure 5D and the growth curves in S5 show that gentamycin exhibits similar bistable yet complex dynamics for 1.5≤A≤4 µM.

Figure 5E,F and the growth curves in S5 show the bistable complex dynamics obtained with the two β-lactam antibiotics ampicillin and carbenicillin. These antibiotics prevent synthesis of new cell-wall peptidoglycan and consequently cause cell-lysis—bacteriolytic. With ampicillin, the bistable range for initial loads of 10–10^6^ CFU/mL is 8≤A≤12 µM (Figure 5D) whereas for carbenicillin a bistable range of 8≤A≤16 µM is observed (Figure 5E). These findings prove that both BAD and BIK mathematical models cannot describe the bacterial dynamics with ampicillin or carbenicillin.

### 4.5. The Existence of a Bistable Range for Inoculums 10–10^6^ CFU/mL Indicates That the MIC for Inoculums Larger than 10^6^ CFU/mL Is Higher than the Standard bMIC

The experiments described in Figure 3 and Figure 4 identify the bistable concentration ranges for initial inoculums of 10–10^6^ CFU/mL, namely, at or below the standard inoculums used for determining the bMIC. Theoretically, the observed trend of inoculum-increasing MIC at low inoculums should continue to larger inoculums—it stops only when the MIC reaches the concentration Ae of Figure 3E and 1 respectively in the BIK and the BAD models, and, similarly, when A is increased past the bistable range of a more complex BFMD model. Indeed, Figure 6A shows that a concentration of 200 µM ampicillin does not overcome an initial inoculum of 10^8^ CFU/mL whereas doubling this concentration to 400 µM does. It follows that the MIC for 10^8^ CFU/mL is in the 200–400 µM range, namely, Ae, the highest concentration at which bistability occurs with ampicillin, is beyond 200 µM (order of magnitude higher than the bMIC 12–14 µM, see Table 1 and [12].

### 4.6. The Cause for the Complex Dynamics Is not Necessarily the Antibiotics Stability

The BMFD dynamics is associated with factors that change during the experiment and influence the bacterial growth. The rich medium we use guarantees that nutrients are not the limiting factor (Figure 2). Naturally, the next factor to test is the antibiotics stability during the course of the experiment (Figure 7A). We find that after 24 h, ampicillin potency decreases (MIC doubles) due to exposure to high bacterial loads whereas carbenicillin potency increases due to its exposure to the medium (Figure 7A right panel). Chloramphenicol, which has simple BAD dynamics is stable for at least 24 h, whereas tetracycline, which also has simple BAD dynamics is less effective after 24 h, independently of bacterial presence. MIC experiments with oxacillin, which is a beta-lactam antibiotic resistant to beta-lactamase degradation, demonstrate that even stable antibiotics can produce BMFD bistable dynamics (Figure 7B–E).

### 4.7. Bistability Occurs with Diverse Biochemical Mechanisms of Action

Table 1 summarizes the mechanism of action and the type of the antimicrobial agent we tested. Bistability is detected for all these distinct types of agents.

## 5. Discussion

In all agents we tested, including all cationic antimicrobial peptides and all conventional antibiotics, independently of their biochemical mechanism of action (Table 1), we found the inoculum effect. At a certain range of concentrations, which is specific for every drug and experimental setting, the system exhibits a bistable behavior in which large loads survive and small loads are inhibited. We identified three distinct types of bistable dynamics, as summarized in flow chart 4, and developed an experimental framework that enables to determine which type of dynamics is realized for a given antimicrobial agent. In particular, for certain cases (all the cationic antimicrobial peptides and all the commercial bacteriostatic antibiotics we tested), we established experimentally that simple mathematical models (BIK, Equation (2) and BAD, Equation (3), respectively) are adequate for describing the bistable growth dynamics.



**Flow Chart 4. The division to three types of bistable dynamics.**

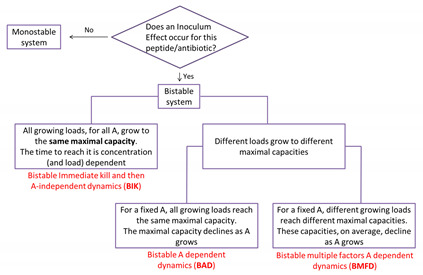




**The CAM-induced IE:** We found that all the tested peptides (polymixin B, K6L9, MSI, and melittin) lead to a bistable bacterial growth, demonstrating for the first time that antimicrobial peptides induce an inoculum effect (Figure 3 and Appendix A in the supplement).

Because antimicrobial peptides kill bacteria in a mechanical fashion by attaching to the bacterial membrane via electrostatic forces and formation of pores/lesions through which the bacterial content flows out, they cause almost immediate death [40]. After initial killing by the peptide there are two possible outcomes: Either all bacteria are eradicated, or some live bacteria are left. In the latter case, the live bacteria grow normally as the control, independently of the antimicrobial peptide that was primarily present in the medium (see our control experiments in Appendix A).

The initial “decision” of whether to go extinct or survive the peptide is highly dependent on the initial bacterial concentration present in the well, and therefore, an inoculum effect occurs. Mechanistically, this model is supported by various biophysical studies done by others and us, showing that CAMPs bind and kill both Gram-negative and positive bacteria within minutes [41,42] see Appendix A. The consequence of this unusual interaction is mathematically described by our BIK model, Equation (2)—there is a density-dependent and concentration-dependent immediate kill function, with the surviving bacteria growing exactly as the control, obeying a deterministic one-dimensional ordinary differential equation with a growth function that is found experimentally (Figure 3B).

It was recently established that when cell-penetrating peptides are employed in conjunction with AMPs, the MIC is reduced [43,44,45]. We do expect, as we established with all other non-linear antimicrobial mechanisms of action, that IE will be identified also when such combinations are employed. Whether or not the surviving bacteria will grow as the control does (namely exhibiting BIK dynamics), or, perhaps, slower due to the cell-penetrating peptides (BAD or BMFD dynamics), is an open question for future studies.

**The complexity of IE induced by antibiotics:** We provide an experimental methodology to examine when the growth dynamics is adequately described by a simple one-dimensional model or when higher-dimensional models must be employed.

For the two tested commercial bacteriostatic antibiotics that target the ribosome (tetracycline and chloramphenicol), for a fixed antibiotics concentration, all duplicates, and all different initial loads that grow collapse to one growth function (Figure 4 and Appendix A). We conclude that in these cases, when growth occurs, stochastic effects, antibiotics stability, and other limiting factors are insignificant. Thus, the deterministic one-dimensional ordinary differential equation BAD model (Equation (3)) adequately describes the dynamics, with growth functions depending on the initial antibiotics’ concentration [A_0_]. We found these functions from the experimental data and observed that they might have multiple maxima for large [A_0_] (Figure 4 and Appendix A). This can be indicative of a change in the collective behavior of bacteria in stress at higher densities and needs to be further studied. These experimentally derived functions may be utilized in future mathematical models of bacterial growth dynamics such as was done in [11,12,18,35].

Bacterial growth with beta-lactam antibiotics, which specifically target the cell wall, or with aminoglycosides that inhibit the ribosome, cannot be described by a single time independent ordinary differential equation: *dB/dt* is not a function of *B(t)*. Indeed, we observe that different loads grow differently and even duplicates have different growth curves and different maximal capacity (Figure 5 and Appendix A). In some of these cases, the dynamics appears to shift from 1d dynamics in the lower antibiotic concentrations towards at least 2d dynamics for high concentrations (Figure 5C–F shows that the maximal growth capacity of different loads collapses to one curve for low A but not for large A). Interestingly, the transition seems to occur in concert with the appearance of the bistability.

To describe this behavior, one needs to construct the more complex BMFD mathematical models that include additional variables and time-dependent factors. Equations (2) and (3) provide the simplest type of such models (see [11,18] for a variety of such models).

**The role of experimental conditions:** While growth in rich medium is well described by the traditional limited growth mathematical model (1), minimal medium leads to complex behavior even without the introduction of anti-microbial agents. To eliminate this variability in our experiments, we conducted all experiments with antimicrobials in the rich medium setup and found a bistable range in all of these experiments. Mathematical models show that if one decreases the growth rate at the exponential phase of a one-dimensional BAD model, bistability is suppressed and monostability arises [20,36]. An experimental set up that realizes such conditions (e.g., by a chemostat) and its implications on best strategies for treatment protocols are left for future studies.

**The IE and MIC:** We concentrated on establishing the IE for low inoculums, below the standard loads for bMIC. Importantly, an increase in the density dependent MIC with the initial inoculum (as in Figure 1B) implies bistability whereas a decrease in the density dependent MIC with the inoculum (as in Figure 1A) is associated with monostability. The mathematical theory shows that bistability typically implies that the standard bMIC is insufficient for fighting inoculums that are larger than the standard inoculum of 10^6^ CFU/mL. Figure 6 demonstrates that this is indeed the case, corroborating similar findings in [11,12,18].

**Mechanistic explanation of IE:** While IE was identified in previous studies of particular conventional antibiotic agents and bacteria [10,11,12,13,14,18], previous explanations of its appearance included genetic and/or phenotypic population heterogeneity and additional factors that vary during the experiments. These were modelled, for example, by deterministic multi-dimensional equations of classical reaction kinetics [11,18] or by two-dimensional PK (Pharmakokinetics)/PD (Pharmakodynamics) dynamics [12]. Here we show that for some cases (the bacteriostatic antibiotics) the one-dimensional BAD model can explain the resulting growth curves by density-dependent mechanisms alone. By Ockham’s razor principle, we assert that the BAD models are adequate for these cases: The more complex reasoning can be neglected for explaining bacterial growth in rich media under bacteriostatic antibiotics.

On the other hand, we demonstrated that in other cases (bistable bacterial growth with bactericidal and bacteriolytic antibiotics), higher dimensional models are required. These could possibly be described by either density-dependent mechanisms (such as the production of β-lactamase or, for other bacteria, by extracellular PH variations [12]) or models involving population heterogeneity, see [11,12,18] and references therein. For example, it is well established that *E. coli* MG1655 produces β-lactamase, an enzyme that hydrolyses β-lactam antibiotics such as ampicillin and carbenicillin [11,38]. If the antibiotic is being inactivated by the bacteria, higher initial loads produce larger concentrations of the enzyme over the course of time, and therefore, the antibiotic concentration will decline faster. Our experimental results with oxacillin demonstrate that even when the β-lactam is not degraded by the bacteria and the drug concentration remains constant over time, IE appears, and the growth dynamic may be still multi-dimensional. Interestingly, these results also exhibit high sensitivity to the experimental settings. This sensitivity could be possibly attributed to the stress induced by the very high antibiotic concentrations (*E. coli* are often naturally resistant to oxacillin [46]), possibly introducing an additional time and antibiotic concentration-dependent factor to the system. Additional experimental work in which these factors are monitored is needed for clarifying the dominant mechanisms involved in the more complex settings. While the identification of mechanisms that control the IE may be pivotal for developing more efficient antimicrobial drugs, we assert that such an identification is inessential and is not always helpful in predicting the bacterial growth dynamics.

**Hypothetical therapeutic implications:** From a therapeutic point of view, understanding the particular bacterial growth dynamic in the presence of different classes of antimicrobials and starting from various bacterial inoculums is central for the assignment of the correct treatment for various bacterial infections (see [11,12,18]). When early diagnosis and treatment are impossible, the ability to predict bacterial growth dynamics in the presence of a selected treatment becomes indispensable. In such cases, the estimation of bacterial load present in the infection site, and the knowledge of the type of growth dynamic of the infecting bacterium with different antimicrobials, would allow for a personalized treatment in terms of dosage and frequency of treatment [11,12,18,35,47].

Models of in-vivo dynamics usually require a BMFD model since the antibiotics clears from the body and the nutrients supply varies. Once an in-vitro BAD model is established, the form of the function $F_{BMFD}\left(B,A\right)$ can be identified with the experimentally derived $F_{BAD}\left(B,A\right).\;$The study of in-vivo bacterial infections requires additional considerations, such as the presence of the immune system, the clearing of the drugs, as well as environmental conditions that are not only subjected to changes by the bacteria but also by the host (e.g., the host may control limiting factors for the bacterial growth, and such factors invoke higher dimensional dynamics, see Figure 2C,D). Nevertheless, studies have shown [17,48] that quite often, in-vitro results regarding interaction dynamics between bacteria and antimicrobials are indicative of such dynamics in-vivo and may be used as building blocks for the in-vivo models [36].

**Outlook:** Our current efforts are concentrated on simulating the results based on the fitting of our experimental data and studying their implications. To achieve a successful clinical treatment, basic growth dynamics rules such as those presented herein should be adapted to include additional parameters such as nutrient limiting factors, drug clearance, the action of the immune system, and the level of drug resistance of the specific bacterial species present at the infection site. Inclusion of these additional factors into the bistability of the basic bacteria-antimicrobial agents’ system can shed light on the relation between in-vitro and in-vivo growth dynamics.

## Figures and Tables

**Figure 1 antibiotics-10-00087-f001:**
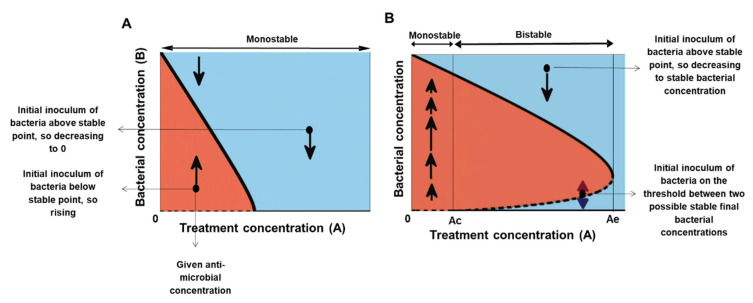
**One dimensional models of monostable and bistable bacterial growth in the presence of antimicrobial agents.** The *x*-axis represents the collection of antimicrobial concentrations, the *y*-axis shows the bacterial amount. In this plot, the bacterial dynamics correspond to motion along vertical lines—from any initial inoculum at a given antimicrobial concentration, the bacterial load increases (moves upward) in the red regions and decreases (moves downward) in the blue regions. The solid/dashed lines correspond to stable/unstable equilibrium states, respectively. (**A**) The monostable model has a single stable equilibrium bacterial concentration for any antimicrobial concentration (a thick full line). (**B**) The bistable model has a range of concentrations, the bistable range, where the system has at least two possible stable equilibria (two full lines). The upper solid line represents the antimicrobial-concentration dependent maximal capacities. The dashed line of unstable equilibria represents the collection of critical bacterial loads $B_{c}\left(A\right)$ above which the bacteria grow to the upper equilibrium branch and below which the bacteria become extinct or inhibited. A_c_ is a minimal antimicrobial agent concentration that inhibits/kills a minute number of cells. A_e_ is a concentration that inhibits/kills every tested bacterial load. The bistable range corresponds to concentrations between A_c_ and A_e_.

**Figure 2 antibiotics-10-00087-f002:**
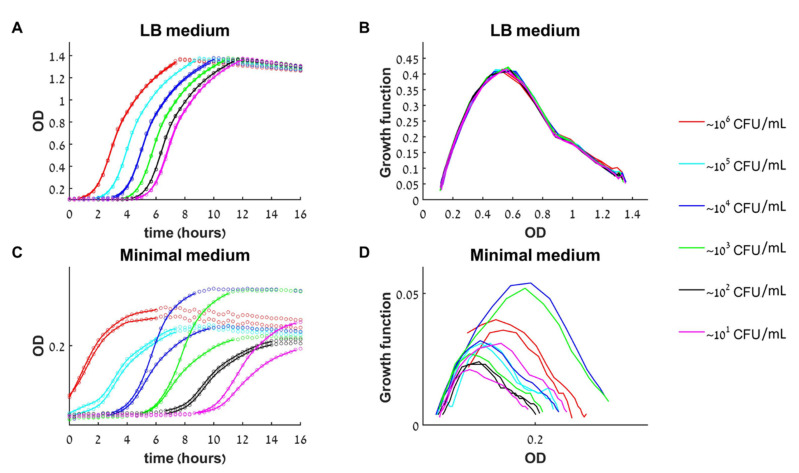
**Growth dynamics of untreated bacteria in rich and minimal medium.** (**A**) The growth of 12 initial loads (six different loads in duplicated) in rich medium (LB) as monitored by a change in optical density (OD) in time. Circles—data points, solid line—smooth approximation to data used for calculating the growth function, see methods. (**B**) The growth curves [experimental d*OD*/dt versus OD] collapse to a single growth function. (**C**) In minimal medium, there is high variability between duplicates and a strong influence of the initial load on the growth rate and the maximal population size. (**D**) The different growth curves in minimal medium do not collapse to a single growth function.

**Figure 3 antibiotics-10-00087-f003:**
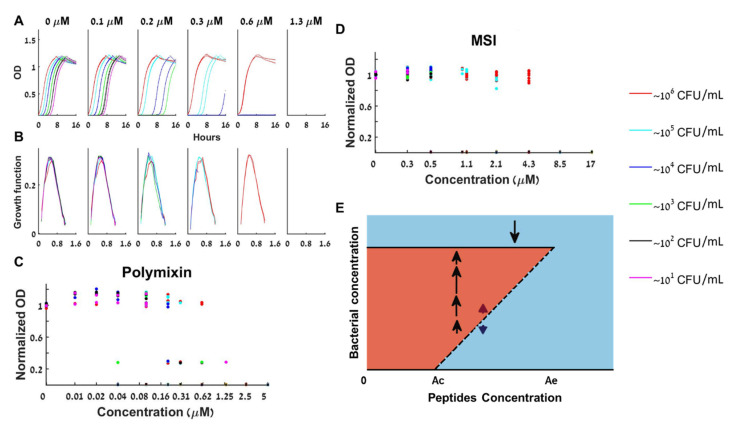
**Bacterial growth dynamics under antimicrobial peptides treatment.** (**A**) At the bistable range, a given bacterial load is either extinct or grows to full capacity as shown for polymixin B. (**B**) The specific growth functions are independent of both the loads and the peptide concentration. (**C**) The normalized maximal growth capacity of each growing load is independent of the polymixin B concentration—three repeats together. (**D**) Normalized maximal growth capacities for all loads and peptide concentrations for MSI-78 (Pexiganin) show similar bistable behavior—4 repeats together. (**E**) The peptide–bacteria interaction can be described by a bifurcation diagram in which bistable behavior occurs between Ac and Ae. See also separate experimental results in Appendix A.

**Figure 4 antibiotics-10-00087-f004:**
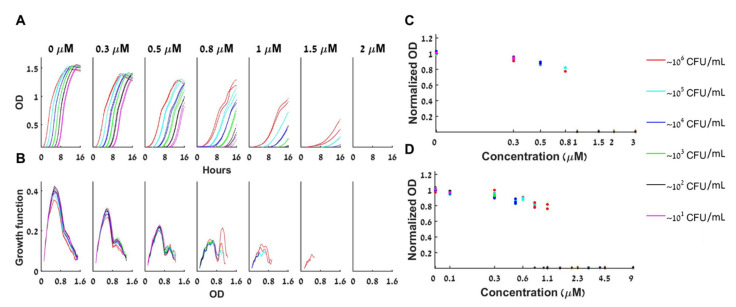
**Dynamics of bacterial growth with Tetracycline.** (**A**) The bacterial population is either extinct or growing with a similar growth curve under a given concentration of tetracycline. At any given antibiotic concentration A (one of the seven subplots), all loads that grow have similar growth curves, shifted in time (different colors correspond to different initial loads as indicated, duplicates are shown). For A > 1 µm some of the small loads fail to grow altogether. The maximal growth capacity declines as the antibiotic concentration A rises. (**B**) The specific growth function (*dOD/dt* versus OD) of the different loads changes as a function of *A* but remains fairly constant for all loads at a given *A*. (**C**) The maximal capacity of all loads at a given tetracycline concentration A is always similar and declines as A becomes bigger. Normalized maximal capacities of one representative experiment are presented. (**D**) The normalized maximal capacity for six experiments together exhibits bistability and the same decline with the concentration A.

**Figure 5 antibiotics-10-00087-f005:**
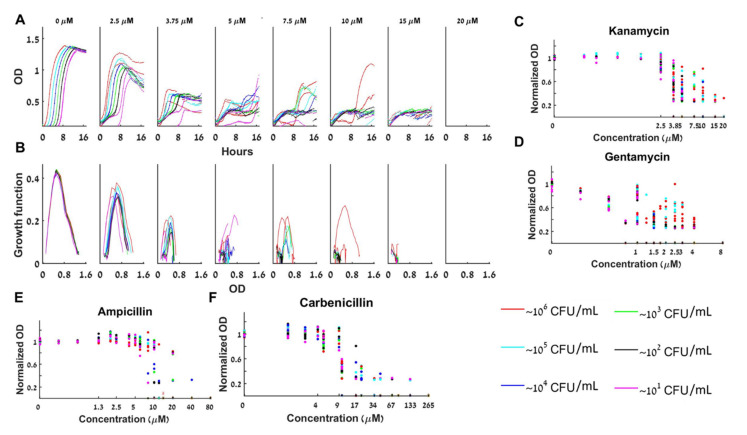
**Bacteria grown with β-lactam or aminoglycoside antibiotics exhibit complex bistable dynamics.** (**A**) The growth of different initial loads for a fixed initial concentration of kanamycin has different forms and thus their growth curves do not collapse to a single growth function (**B**). Collective results of normalized maximal capacity are shown for (**C**) six repeats with kanamycin. (**D**) Four repeats with gentamycin. (**E**) Four repeats with ampicillin. (**F**) Four repeats with carbenicillin. The maximal capacities are load and dose dependent. See also separate experimental results in Appendix A.

**Figure 6 antibiotics-10-00087-f006:**
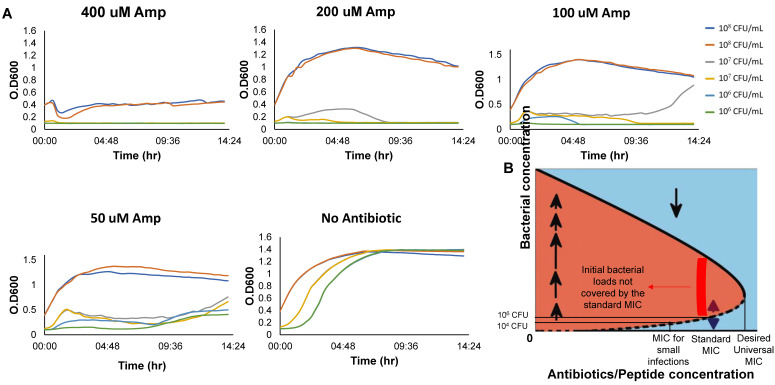
**Growth of Inoculums higher than the standard with Ampicillin.** (**A**) Growth curves of different initial inoculums for 14 h in high concentrations of Ampicillin. (**B**) The standard MIC and its relation to MICs of other inoculums for bistable growth dynamics.

**Figure 7 antibiotics-10-00087-f007:**
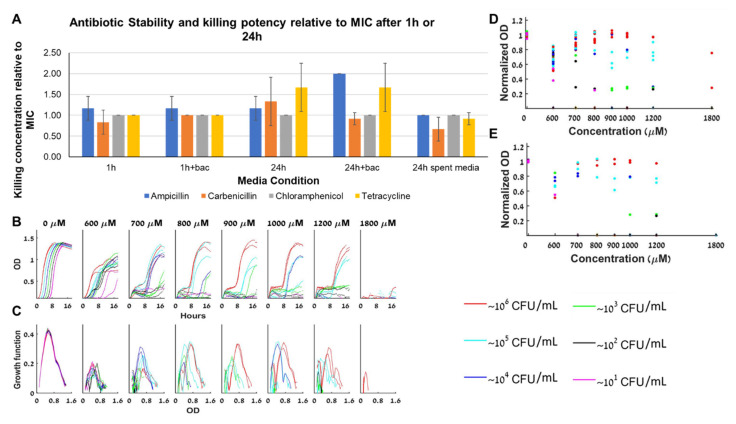
**Antibiotic stability.** (**A**) Minimal inhibitory concentration of the antibiotic relative to the MIC as a function of several media conditions (incubation with or without antibiotics and bacteria as indicated). (**B**) Different loads of bacteria grown with oxacillin have a similar growth slope in the exponential growth phase at almost all A’s (except for loads that fail to grow altogether). (**C**) The specific growth function of the different loads in oxacillin changes with the loads and with A, with no obvious trend with respect to A. In particular, the peak of the function remains essentially constant at different oxacillin concentrations. (**D**) The maximal capacity at all oxacillin concentrations is always similar, and does not decline as A becomes bigger, although different loads reach different maximal capacities in each antibiotic concentration. (**E**) The maximal capacities of different loads grown in oxacillin from 6 different experiments. See also separate experimental results in Appendix A.

**Table 1 antibiotics-10-00087-t001:** Antibacterial agent properties and tested bMIC ranges concentrations.

Agent	Type	Mechanism	Target	MIC (µM)
Chloramphenicol	Chloramphenicol	Bacteriostatic	Ribosomal 50S	5–20
Tetracycline	Polyketide	Bacteriostatic	Ribosomal 30S	2–4.5
Gentamycin	Aminoglycoside	Bactericidal	Ribosomal 30S	4–8
Kanamycin	Aminoglycoside	Bactericidal	Ribosomal 30S	10–20
Ampicillin ^a^	β-lactam	Bacteriolytic	Cell-wall PBP	12–14
Carbenicillin ^a^	β-lactam	Bacteriolytic	Cell-wall PBP	21–43
Oxacillin	β-lactam ^b^	Bacteriolytic	Cell-wall PBP	1800
Polymixin B	CAMP	Bactericidal	Membrane	0.3–1.25
Melittin	CAMP	Bactericidal	Membrane	20
MSI	CAMP	Bactericidal	Membrane	17
K_6_L_9_	CAMP	Bactericidal	Membrane	10–20

^a^*E. coli* K12 MG1655 is a β-lactamase producer that can cleave β-lactam antibiotics. PBP is a penicillin-binding-protein. ^b^ Oxacillin is a β-lactamase resistant β-lactam.

## Data Availability

Figures in supplementary material present available data. Data is available and can be accessed directly from the authors.

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
