# Peer review of "Bistable Bacterial Growth Dynamics in the Presence of Antimicrobial Agents"

_antibiotics, 2021, doi:10.3390/antibiotics10010087_

Round 1

Reviewer 1 Report

The manuscript on bacterial growth dynamics in presence of different antibiotics shares interesting relationship between both and how peptides in specific shows inoculum effect in E.coli cell population. In this era of rising antimicrobial resistance a study that focuses on the IE frame work of bacterial population in response antibiotics might help in controlling rise in MDR variants to certain level by administering more accurate amount of antibiotics in different aspect,

Minor comments:

1)Please use a specific pattern for citing reference: Line 592

2)Check line 41

3)Restructure line 524-525

Author Response

We thank the reviewer for finding our work interesting and for finding it useful in today’s world.

1) Please use a specific pattern for citing reference: Line 592

Thank you for this comment, the citing pattern was changed here to fit the rest of the manuscript.

2) Check line 41

Thank you for this comment, corrected.

3) Restructure line 524-525

Thank you for this comment, we broke the long sentence to two parts

Reviewer 2 Report

The manuscript "Bistable bacterial growth dynamics in the presence of antimicrobial agents" by Frenkel et al talks about the inoculum effect. Although the IE effect has been known for quite some time, the authors have very well described it using mathematical derivations and equations.

The work itself is interesting and meaningful and is making a good connection between antibiotics and AMPs as far as inoculum effect is concerned.

However, there are some minor improvements that could be made before it is accepted in the Antibiotics journal.

1)The experimental materials and methods need improvement. Please capitalize Ampicillin in Line 137 as this is the first word of the sentence. Please write Gentamicin instead of gentamycin in Line 138.

2) Is Fluka BioChemika a single company from where you purchased Polymixin B. Also, I am not sure whether reference 16 is placed in the right context. If possible, try to improve the quality of the graphs. Some of the lines are hard to see in Figure 4 a and b

3) Likewise, capitalize Initially in Line 151, A in Line 164, CFU in 167, etc.

4) The authors have taken the membrane-targeting antimicrobial peptides for their study. Can they talk a bit about cell-penetrating peptides also so that their study could have a higher generalization? 

Author Response

We thank the reviewer’s interest in our work and happy to see the reviewer found it meaningful.

1) The experimental materials and methods need improvement. Please capitalize Ampicillin in Line 137 as this is the first word of the sentence. Please write Gentamicin instead of gentamycin in Line 138.

We thank the reviewer for suggesting improvement in the materials and methods section and made all suggested changes – all capitalizations and spelling corrections where performed as asked for.

2) Is Fluka BioChemika a single company from where you purchased Polymixin B?

Yes, Polymixin B was purchased only from Fluka BioChemika.

3) I am not sure whether reference 16 is placed in the right context.

Thank you so much for noticing, indeed – this reference was misplaced and now deleted from the experimental materials and methods section. 

4) If possible, try to improve the quality of the graphs. Some of the lines are hard to see in Figure 4 a and b

Thank you for this important comment. All figures in the paper (including figure 4) were now re-made and their quality improved.

5) Likewise, capitalize Initially in Line 151, A in Line 164, CFU in 167, etc.

All indicated words throughout the manuscript were capitalized, thank you for noticing these unfortunate mistakes.

6) The authors have taken the membrane-targeting antimicrobial peptides for their study. Can they talk a bit about cell-penetrating peptides also so that their study could have a higher generalization? 

Thank you for this important comment regarding generalization of the study, this topic is now shortly discussed in the discussion section (see lines 514-519) and some relevant references to CPPs were added (references [43-45]).

Reviewer 3 Report

The manuscript is clearly written in good English. The design of the experiments and their interpretation are correct. As abstract objects, bistable systems are very interesting from a mathematical point of view. Describing bacterial growth, such models provide a possibility to find the optimal parameters, which usually cannot be localized using simple models. Thus, the presented results are important from a practical point of view too.  

I recommend to accept the paper in present form. But to make it better, some "mathematical rigor" could be added. For example, in the lines 233-235 the authors say: "If a unique curve is found when this procedure is repeated for several initial inoculum ?(0), then model (1) is adequate for describing the control bacterial dynamics. Otherwise, a more complex model is needed." Of course, intuitively it is clear, but when we are talking about a mathematical apparatus, we cannot say just "a unique curve", we have to provide a criteria to distinguish the curves.   

Author Response

We thank the reviewer for finding our work helpful and practical as well.

I recommend to accept the paper in present form. But to make it better, some "mathematical rigor" could be added. For example, in the lines 233-235 the authors say: "If a unique curve is found when this procedure is repeated for several initial inoculum (0), then model (1) is adequate for describing the control bacterial dynamics. Otherwise, a more complex model is needed." Of course, intuitively it is clear, but when we are talking about a mathematical apparatus, we cannot say just "a unique curve", we have to provide a criteria to distinguish the curves.   

We thank the reviewer for this comment. We have now made the following more precise statement (lines 233-237): “If, to a good enough approximation, a unique curve is found when this procedure is repeated for several initial inoculum, then model (1) is adequate for describing the control bacterial dynamics. Otherwise, a more complex model is needed. A threshold condition on the deviations of the experimental   curve from its mean can be set to distinguish between these behaviors.